# TBG-Driven Minimization of Noise-Resistant Adaptive Sharpness Awareness

## Abstract

Driven by the sharpness of the loss surface effectively indicate the generalization gap, sharpness-awareness minimization (SAM) aims at flat minima within the loss landscape. However, to protect sensitive information and enhance privacy security, noise will be added to the model, which inevitably degrades the model's generalization performance. In this paper, we introduce the time-base generator (TBG) based on discrete systems and provide a boundedness theorem for discrete systems. On this basis, we propose a noise-resistant adaptive sharpness-awareness minimization method (NRASAM), which suppresses noise through gradient decay and historical gradient integration. Furthermore, we utilized the TBG theory to adjust the algorithm parameters, resulting in the TBG-NRASAM algorithm. We provide a rigorous theoretical analysis that confirms the convergence and noise resistance of the proposed method under noisy conditions. Extensive experiments across multiple architectures and benchmarks demonstrate that our approach consistently improves generalization and stability compared to existing SAM-based methods.

## 1 Introduction

In recent years, deep learning technologies have achieved breakthrough prog-ress in various fields, demonstrating powerful representational and reasoning capabilities, particularly in complex tasks such as computer vision (Wang et al. (2025); Malik et al. (2025)) and natural language processing (Spangher et al. (2025); Sun et al. (2024)). However, as model scales continue to expand and training data becomes increasingly abundant (Maini et al. (2024)), privacy leakage and data security issues (Fang et al. (2024)) have gradually become critical factors limiting the practical deployment of these technologies. To protect users' sensitive information, a common approach is to introduce random noise during the training process to achieve differential privacy protection, such as adding perturbations during gradient updates or data input stages (Wang et al. (2024); Xu et al. (2019)). Although such methods effectively reduce privacy risks, the introduction of noise often interferes with the model optimization trajectory (Jayaraman & Evans (2019)), leading to degraded generalization performance and weakened robustness.

Even in ideal training environments, improving the generalization performance of deep neural networks is challenging (Zhang et al. (2021)). Their loss function surfaces often contain numerous sharp minima (Hochreiter & Schmidhuber (1994); Dinh et al. (2017); Wei & Ma (2020)), making the models highly sensitive to input perturbations. Sharpness-Aware Minimization (SAM) explicitly optimizes the flatness of the loss landscape and demonstrates excellent generalization capabilities on clean data (Foret et al. (2020)). Nowadays, there are already many relevant studies based on SAM (Liu et al. (2022); Wang et al. (2023)). In particular, Adaptive Sharpness-Aware Minimization (ASAM) effectively addresses the sensitivity issue of traditional sharpness under parameter rescaling by introducing adaptive sharpness, a scale-invariant generalization measure (Kwon et al. (2021)). Curvature Regularized Sharpness-Aware Minimization (CR-SAM) addresses the issue of high non-linearity in the loss landscape by introducing a normalized Hessian trace to accurately measure the curvature of the loss landscape and integrating it as a regularizer into SAM training (Wu et al. (2024)).

However, when noise is present during training, the model's generalization ability will be inevitably reduced (Phan et al. (2016)). Accordingly, noise robustness has now become a vibrant research

area (Ren et al. (2021); Mansour & Heckel (2023)). In fact, gradient descent methods that are influenced by historical gradients often yield better performance. For example, momentum methods (Qian (1999)) accelerate convergence, and the classic PID method (Borase et al. (2021)) is another notable example. Additionally, motivated by a denoising neural algorithm used for solving time-varying linear equations (Jin et al. (2018)), we introduce a gradient history integral term with decay characteristics and an adaptive adjustment mechanism to suppresses the interference of noise on parameter updates.

This paper draws on stability theory from dynamic systems and proposes a TBG based on discrete-time systems, establishing a novel boundedness theoretical framework. Based on the above discussion, we design the noise-resistant adaptive SAM algorithem (NRASAM). And we further propose a TBG-driven noise-resistant adaptive SAM method (TBG-NRASAM) to enhance the model's generalization capability under noisy conditions. The main contributions of this paper are as follows:

- A TBG based on discrete systems and its boundedness theory are proposed, providing a new tool for stability analysis of non-exponentially decaying systems.

- A noise-resistant optimization framework with rigorous theoretical guarantees is constructed, which effectively suppress noise interference.

- The superiority and robustness of the proposed method are validated through multiple practical tasks.

The structure of this paper is arranged as follows. In Section 2, we briefly introduce the relevant preliminary knowledge, including the SAM, ASAM algorithms and related noise resistance methods. In Section 3, the related TBG definitions and theories are proposed. In Section 4, we present the noise-resistant adaptive SAM algorithm and provide the corresponding theoretical convergence argument. In Section 5, we demonstrate the performance of NRASAM and TBG-NRASAM on some models and datasets. Section 6 will provide a summary discussion.

## 2 SHARPNESS-AWARE MINIMIZATION

The model's performance on the training set and test set will have a generalization gap, as shown in Figure 1. A small generalization gap indicates that the model's performance on the training set and test set is not significantly different, which means that the model has strong generalization performance. The loss curve on the right side of Figure 1 is flatter than that on the left side, and its generalization gap is also smaller. The SAM algorithm is an algorithm that adds a sharpness penalty term to the optimization objective to simultaneously minimize the loss value and loss sharpness in order to find the minimum value in the flat landscape.

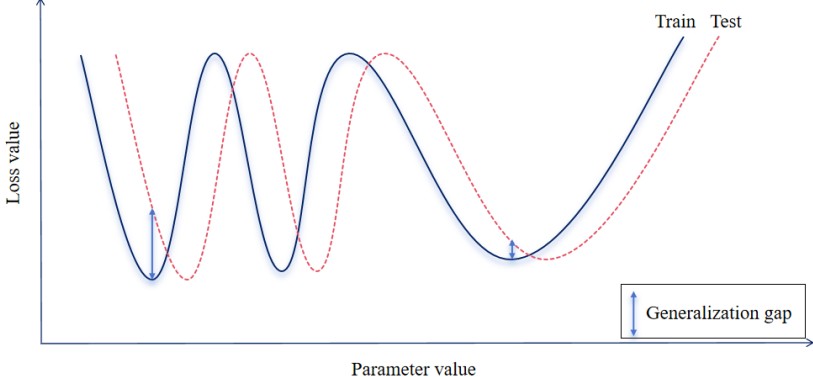

Figure 1: Generalization gaps of train dataset and test dataset.

SAM aims to optimize the following Min-Max problem:

$$\min_w \left[ \max_{\|\epsilon\|_p \leq \rho} L_{D_{\text{train}}}(w + \epsilon) + \frac{\lambda}{2}\|w\|_2^2 \right],$$

where $\rho > 0$ is the perturbation radius, $\lambda$ is the weight decay coefficient, and $p \geq 1$ specifies the norm type. This objective is solved through an iterative two-step process (Foret et al. (2020)):

$$\begin{cases} \epsilon_t = \rho \cdot \frac{\nabla_w L_{D_{\text{train}}}(w_t)}{\|\nabla_w L_{D_{\text{train}}}(w_t)\|_2} \\ w_{t+1} = w_t - \alpha_t \left( \nabla_w L_{D_{\text{train}}}(w_t + \epsilon_t) + \lambda w_t \right). \end{cases}$$

The first step employs the gradient direction as the perturbation direction because the gradient direction is the direction in which the loss increases most rapidly. The magnitude of the perturbation is controlled by $\rho$, which indicates searching for the worst-case perturbation within an Euclidean ball. Namely, the objective of the first step is to seek the perturbation point $w_t + \epsilon_t$ that maximizes the loss within a fixed radius $\rho$ neighborhood. The second step is to compute the gradient at the perturbed point $w_t + \epsilon_t$, rather than at the original parameter point. This kind of update enable the parameters to escape from sharp regions and move towards flatter regions.

## 3 TIME-BASE GENERATOR

### 3.1 SOME DEFINITIONS

This section first defines the TBG based on discrete systems.

**Definition 1.** *Note that positive function $\mu(t, t_0)$ satisfy:*
*$\mathcal{A}1)$ $\lim_{t \to +\infty} \mu(t, t_0) = 0$, $\mu(t, t_0)$ is decreasing with respect to $t$;*
*$\mathcal{A}2)$ Select proper positive constant $\alpha$ and $K$ such that $\lim_{t \to t_p} K\mu^\alpha(t, 0) \leq \epsilon$, where $\epsilon$ is a small positive constant that can be arbitrarily chosen;*
*$\mathcal{A}3)$ $\mu(t, c)\mu(c, t_0) = \mu(t, t_0)$ (c is a positive constant), and $\mu(t, t-1) \leq b < 1$.*

The TBG is defined by a positive, decreasing function $\mu(t, t_0)$ that decays over time, where $t_0$ represents the initial moment.

**Remark 1.** *In addition to exponential-type TBGs, there are many other types of TBGs available for selection. For instance, $\mu(t, t_0) = \frac{1}{(1+t-t_0)^p}$, where $p > 0$ is a constant; $\mu(t, t_0) = \frac{\ln(e+t_0)}{\ln(e+t)}$, where $t \geq t_0 \geq 0$; $\mu(t, t_0) = \frac{\cosh(at_0)}{\cosh(at)}$, where $a > 0$, $t \geq t_0 \geq 0$, and so on.*

We first consider the linear system $x_m = A_{m-1}x_{m-1}$, where $x_m$ belongs to the $m$-dimensional real space, and $A$ is an $m \times m$ matrix. In the discrete system formulation, $x_m$ is the state vector at step $m$, $A_m$ is the state transition matrix. And the compound state transition matrix from step $n$ to $m$ can be defined as:

$$\mathcal{A}(m, n) = \begin{cases} A_{m-1} \cdots A_n, & m > n, \\ I, & m = n. \end{cases}$$

After that, we present Definition 2.

**Definition 2.** *The sequence $(A_m)_{m \in \mathbb{N}}$ is said to admits a $\mu - contraction$ if there exist constants $K$ and $\alpha$ such that*

$$\| \mathcal{A}(m, n) \|_2 \leq K\mu^\alpha(m, n),$$

*where $m \geq n$*

**Remark 2.** *In addition to the TBG based on discrete systems, the application of TBG in continuous systems is extensive and serves multiple purposes. Numerous literature (Becerra et al. (2017); Liu et al. (2023)) indicate that it not only aids in accelerating convergence but also ensures predefined-time convergence while guaranteeing convergence accuracy and stability.*

### 3.2 BOUNDEDNESS THEOREM

Based on the above definitions and analysis, we present the boundedness theorem of the following discrete linear systems. The detailed proof is provided in Appendix A.2.

**Theorem 1.** *If $(A_m)_{m \in \mathbb{N}}$ is a $\mu - contraction$ sequence, then for $x_m = A_{m-1}x_{m-1}$, we have*

$$\| x_m \|_2 \leq K\mu^\alpha(m, n)x_n.$$

In particular, when there is perturbation in the linear system, that is, $x_m = A_{m-1}x_{m-1} + f_{m-1}$, we present the boundedness theory of discrete systems based on the above definitions, which helps to analyze the convergence of the system under perturbation. We provide detailed proof in Appendix A.3.

**Theorem 2.** *If $(A_m)_{m \in \mathbb{N}}$ is a $\mu - contraction$ sequence and $f_l$ is bounded, i.e. there exists $f_{max}$ such that $\| f_l \|_2 \leq f_{\max}$, then for $x_m = A_{m-1}x_{m-1} + f_{m-1}$, we have*

$$\| x_m \|_2 \leq K\mu^\alpha(m, n)x_n + \frac{Kf_{\max}(1 - b^{\alpha(m-n)})}{1 - b^\alpha}.$$

Then, we provide two examples to illustrate the effectiveness of TBG. We consider two cases, as follow.

1. $x_m = \mathrm{e}^{-a}x_{m-1}$,

2. $x_m = \mathrm{e}^{-a}x_{m-1} + 0.1 \cdot \frac{1}{1+e^{-x_{m-1}}}$.

For the first case without perturbation, it can be known that $e^{-a}$ satisfies Definition 1. We plot the cases where $a = 0.1$, $0.5$, and $1.0$ as shown in Figure 2(a). It is not difficult to find that under the effect of TBG, the convergence rate of the curve is faster than that without using TBG. Moreover, the larger the $a$, the faster the convergence rate. For the second case with perturbation, it can be seen from Figure 2(b) that the curve with TBG not only has a faster convergence rate than the one without TBG, but also has a lower convergence lower bound than the one without TBG. In addition, the larger the $a$, the faster the convergence rate and the lower the convergence lower bound.

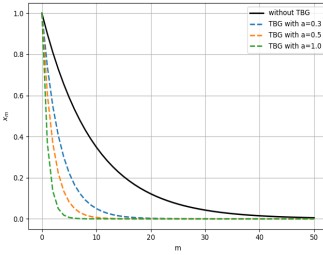
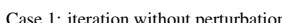
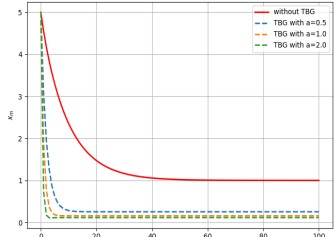

Case 1: iteration without perturbation        Case 2: iterative with sigmoid perturbation

Figure 2: TBG theoretical diagram.

## 4 THEORETICAL ANALYSES

### 4.1 NRASAM ALGORITHM DESIGN

In traditional SAM, perturbations primarily arise from two sources: first, intrinsic perturbations caused by parameter scaling, which make the sharpness of the loss landscape sensitive to parameter scales; second, extrinsic perturbations originating from training data or optimization processes, such as gradient noise or label noise. To address both types of perturbations simultaneously, this section proposes the NRASAM algorithm, which integrates adaptive sharpness minimization with a noise suppression mechanism.

First, to mitigate intrinsic perturbations, the adaptive sharpness measure in Kwon et al. (2021) is used. By applying a normalization operator $T_w$ to correct for parameter scaling, the perturbation region is defined as $\|T_w^{-1}\epsilon\|_p \leq \rho$, ensuring scale-invariant sharpness measurement. The perturbation direction is computed as:

$$\epsilon_t = \rho \frac{T_{w^t}^2 \nabla L_{D_{\text{train}}}(w^t)}{\|T_{w^t} \nabla L_{D_{\text{train}}}(w^t)\|_2}.$$

Second, to suppress extrinsic noise perturbations, a noise-robust update mechanism based on gradient dynamics is incorporated. Inspired by Jin et al. (2018) and Su et al. (2025), we construct a gradient history evolution equation:

$$w_{k+1} = w_k - \gamma h_{k-1} + \delta \sum_{j=0}^{k} h_j,$$

where $\gamma = \eta \cdot \frac{1}{1+\tau\alpha}$, $\delta = \eta \cdot \frac{\tau^2 \beta}{1+\tau\alpha}$. The parameters mentioned in the above equation are derived from:

$$\frac{\mathrm{d}h(w)}{\mathrm{d}t} = -\alpha h(w) - \beta \int_0^t h(w)\mathrm{d}\tau,$$

where $h(w) = \nabla \mathcal{L}_{D_{\text{train}}}(w), \alpha > 0$, $\beta > 0$ are tunable coefficients, $\eta$ is learning rate, and $\tau$ is the sampling interval. Hence we derive the following update rule:

$$\begin{cases} \epsilon_t = \rho \frac{T_{w^t}^2 \nabla L_{D_{\text{train}}}(w^t)}{\|T_{w^t} \nabla L_{D_{\text{train}}}(w^t)\|_2}. \\ w_{t+1} = w_t - \gamma \nabla L_{D_{\text{train}}}(w_{t-1} + \epsilon_{t-1}) + \delta \sum_{i=0}^{t} \nabla L_{D_{\text{train}}}(w_i + \epsilon_i). \end{cases}$$

Let $w_i^{adv} = w_i + \epsilon_i$, the update rule can be reformulated as :

$$\begin{cases} \epsilon_t = \rho \frac{T_{w^t}^2 \nabla L_{D_{\text{train}}}(w^t)}{\|T_{w^t} \nabla L_{D_{\text{train}}}(w^t)\|_2}. \\ w_{t+1} = w_t - \gamma \nabla L_{D_{\text{train}}}(w_{t-1}^{adv}) + \delta \sum_{i=0}^{t} \nabla L_{D_{\text{train}}}(w_i^{adv}). \end{cases} \tag{1}$$

Algorithm 1 illustrates the update process of NRASAM, where $T_w$ mentioned in steps 4 and 5 refers to the adaptive coefficient in ASAM, i.e., the normalization operator, and $\nabla L_B(w)$ represents the loss function gradient with respect to the data set $B$. In particular, the TBG-NRASAM algorithm achieve better convergence by adjusting the values of the parameters $\gamma$ and $\delta$.

---

**Algorithm 1** NRASAM algorithm

---

**Require:** Loss function $l$, training dataset $D := \bigcup_{i=1}^{n}\{(x_i, y_i)\}$, mini-batch size $b$, radius of maximization region $\rho$, learning rate $\gamma$, accumulation coefficient $\delta$, initial weight $w_0$.
**Ensure:** Trained weight $w$
 1: Initialize weight $w := w_0$
 2: **while** not converged **do**
 3:     Sample a mini-batch $B$ of size $b$ from $D$
 4:     $\epsilon_t = \rho \frac{T_w^2 \nabla L_B(w)}{\|T_{w^t} \nabla L_B(w)\|_2}$
 5:     $w_{t+1} = w_t - \gamma \nabla L_B(w_{t-1}^{\text{adv}}) + \delta \sum_{i=0}^{t} \nabla L_B(w_i^{\text{adv}})$
 6: **end while**
 7: **return** $w$

---

## 4.2 CONVERGENCE ANALYSES

Based on the above analyses, we obtain the corresponding boundedness theorem as follows. For the convenience in notation, we let $h_k = \nabla L_{D_{\text{train}}}(w_k^{\text{adv}})$. The detailed proof is provided in Appendix A.4.

**Theorem 3.** *Under the condition of Lemma 2 in A.1, if $\rho < \mu(a, 0)$, where $a$ is the positive constant and $\rho = \max \left\{ \left| \frac{\eta+\gamma+\sqrt{(\eta-\gamma)^2-4\gamma\delta}}{2(\delta+\eta)} \right|, \left| \frac{\eta+\gamma-\sqrt{(\eta-\gamma)^2-4\gamma\delta}}{2(\delta+\eta)} \right| \right\}$, then we have*

$$\lim_{k\to\infty} \|h_k\|_2 \leq \frac{C_\beta M \tau^3}{6\sqrt{2}(1 - \mu(a, 0))}.$$

Taking into account the influence of additive noise $e_k$, the rule for updating (2) is restructured to:

$$\boldsymbol{w}_{k+1} = \boldsymbol{w}_k - \gamma \boldsymbol{h}_{k-1} + \delta \sum_{j=0}^{k} \boldsymbol{h}_j + o_1(\tau^2) + e_k. \tag{2}$$

**Theorem 4.** *Under the condition of Theorem 1, the following inequality holds:*

$$\lim_{k \to \infty} \|h_k\|_2 \le \frac{\sqrt{2}C_\beta}{(1 - \mu(a,0))} \left( \frac{M\tau^3}{24} + \sqrt{2d} \sup_{2 \le i \le k} |c_i| \right).$$

We provide detailed proof in Appendix A.5.

## 5 EXPERIMENTAL RESULTS

In the experiments, we employ the CIFAR10/100 (Krizhevsky et al. (2009))) and SVHN (Netzer et al. (2011)) datasets to evaluate the performance of TBG-NRASAM, NRASAM, NRSAM (Su et al. (2025)), ASAM, SAM, and SGD-M (Sutskever et al. (2013)). Following common practice, the training uses 200 epochs for CIFAR, and SVHN. The momentum and weight decay are set to 0.9 and 0.0001, respectively, with a batch size of 128. A StepLR scheduler is applied to adjust the learning rate at specified intervals (Wei et al. (2023)). For CIFAR and SVHN, the initial learning rate is 0.1, which is decreased by a factor of 0.1 at 20%, 50%, and 80% of the total epochs. The radius parameter $\rho$ in TBG-NRASAM, NRASAM, NRSAM, ASAM and SAM is fixed at 0.1. To simulate real world noise, Gaussian noise sampled from $\mathcal{N}(0, \sigma^2 I)$ is incorporated into gradients (Mumuni & Mumuni (2022)). Models including WideResNet (Zagoruyko & Komodakis (2016)), ResNet (He et al. (2016)), Convmixer (Trockman & Kolter (2022)), MobileNet (Howard et al. (2017)), and Vit-Tiny (Wu et al. (2022)) are evaluated on CIFAR and SVHN. All CIFAR and SVHN experiments are conducted on the RTX 4090 GPU.

### 5.1 EXPERIMENTAL RESULTS UNDER THE INFLUENCE OF ADDITIVE NOISE.

#### 5.1.1 CIFAR-10

We first conducted experiments on the CIFAR-10 dataset. In this part of the experiment, we uniformly set the Gaussian noise parameter $\sigma$ to 0.005. The experimental results are shown in Table 1. Among the five different models, NRASAM performed the best, followed by NRSAM. The accuracy of these two models far exceeded that of the other three optimizers. This indicates that in an additive noise environment, the robustness and generalization performance of the models are effectively enhanced after using noise-resistant methods.

Table 1: Performance of five models with Gaussian noise $\mathcal{N}(0, 0.005)$ on CIFAR-10.

| Model | NRASAM | NRSAM | ASAM | SAM | SGD-M |
|---|---|---|---|---|---|
| Wrn28-2 | **90.45** | 90.30 | 86.81 | 86.72 | 86.54 |
| ResNet32 | **88.75** | 88.27 | 86.15 | 86.02 | 86.15 |
| Convmixer | **78.27** | 78.16 | 74.59 | 75.57 | 75.73 |
| MobileNet | **89.50** | 89.31 | 88.60 | 88.66 | 88.21 |
| ViT Tiny | **75.26** | 75.02 | 67.45 | 66.7 | 64.42 |

We intuitively demonstrated the experimental data using a scatter plot. It can be seen from Figure 3 that the convergence speed of NRASAM and NRSAM in the early stage is much faster than that of the other three optimizers, and they reached high accuracy in a very short number of epochs. In addition, it can also be observed that in the middle and late epochs, the accuracy of NRASAM is always higher than that of NRSAM. All of these demonstrate the prominent advantages of NRASAM in this dataset.

#### 5.1.2 CIFAR-100

We experimented with the performance of four different models on the CIFAR-100 dataset with the Gaussian noise parameter $\sigma$ to 0.005, and the test set accuracy of the experiments is shown in Table 2. It can be seen that the experimental results show a similar trend to that of CIFAR-10. In a noisy environment, among the five optimizers, SGD-M has the worst generalization performance across all models, while NRASAM and NRSAM still outperform the other three optimizers by a

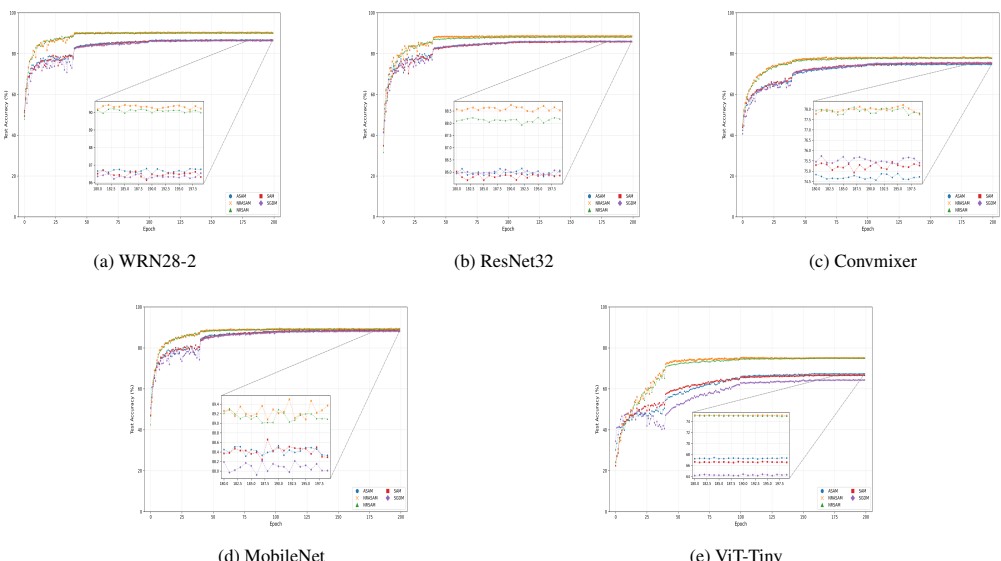

(a) WRN28-2        (b) ResNet32        (c) Convmixer

(d) MobileNet        (e) ViT-Tiny

Figure 3: Performance of five models with Gaussian noise $N(0, 0.005)$ on CIFAR10.

significant margin. Moreover, NRASAM achieves the best performance. As also shown in Figure 4, in the early training epochs, both NRASAM and NRSAM quickly reach high accuracy. In the later stages of training, such as the last 20 epochs, NRASAM almost consistently takes the lead. This is sufficient to illustrate the remarkable efficiency and generalization performance of this optimizer in deep neural networks.

Table 2: Performance of four models with Gaussian noise $\mathcal{N}(0, 0.005)$ on CIFAR100.

| Model | NRASAM | NRSAM | ASAM | SAM | SGD-M |
|---|---|---|---|---|---|
| Wrn28-2 | **67.36** | 66.88 | 60.58 | 60.24 | 60.81 |
| ResNet32 | **63.28** | 62.98 | 59.97 | 58.93 | 59.64 |
| ViT Tiny | **47.53** | 47.41 | 44.18 | 43.91 | 43.05 |
| MobileNet | **66.93** | 66.82 | 66.01 | 66.04 | 65.13 |

### 5.1.3 SVHN

Similarly, we compared the accuracy of five optimizers under five different models with Gaussian noise where $\sigma$ is 0.005. As shown in Table 3, except for Vit Tiny, the accuracy of the other four optimizers is very high. In the environment where noise is added to the gradients, NRASAM outperforms the other four optimizers in all network architectures. Figure 5 shows the convergence process of the test accuracy of these five optimizers. It can be seen that the convergence rate of NRASAM and NRSAM in the early stage is still very prominent, and they quickly reach a high accuracy. Moreover, during the training process, the accuracy of NRASAM is almost always the highest. In summary, NRASAM shows better robustness in a noisy environment.

### 5.1.4 EXPERIMENTAL RESULTS WITH VARIOUS NOISE SCALES

In order to more comprehensively evaluate the performance of NRASAM, in this section, we adjusted the magnitude of the noise. Specifically, we set $\sigma$ to 0.005, 0.008, 0.010, and 0.015 respectively, and tested on the CIFAR-10 dataset using the ResNet32 model. The test results are shown in Table 4. It can be seen that as the noise intensity increases, the accuracy of ASAM, SAM, and SGD-M drops significantly, while the accuracy of NRASAM and NRSAM remains at a high level. Moreover, under high-intensity noise, the advantage of NRASAM becomes increasingly evident.

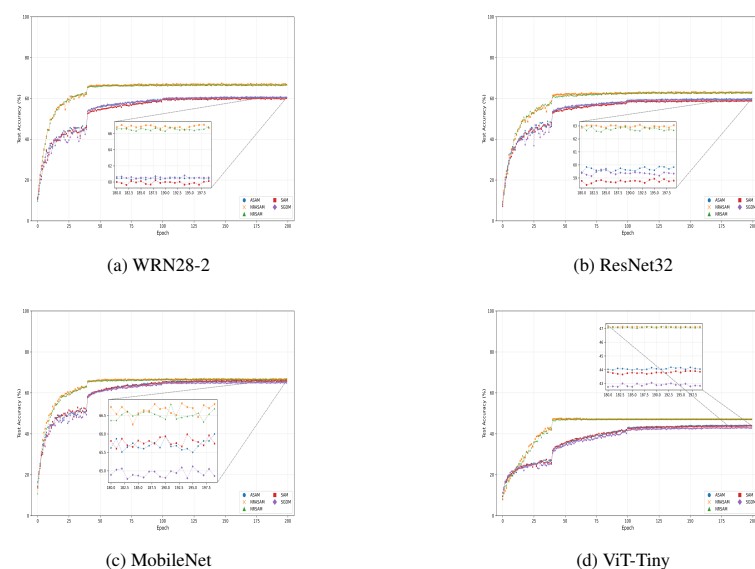

Figure 4: Performance of four models with Gaussian noise $N(0, 0.005)$ on CIFAR100.

Table 3: Performance of five models with Gaussian noise $\mathcal{N}(0, 0.005)$ on SVHN.

| Model | NRASAM | NRSAM | ASAM | SAM | SGD-M |
|---|---|---|---|---|---|
| Wrn28-2 | **95.82** | 95.73 | 95.06 | 95.00 | 95.06 |
| Convmixer | **91.83** | 91.81 | 90.62 | 90.63 | 91.30 |
| ResNet32 | **95.86** | 95.81 | 95.08 | 93.27 | 94.95 |
| ViT Tiny | **87.51** | 86.54 | 66.13 | 72.53 | 69.27 |
| MobileNet | **95.78** | 95.65 | 95.47 | 95.51 | 95.31 |

This fully demonstrates that in a noisy environment, NRASAM still has high robustness and generalization performance.

Table 4: Test accuracy under different noise levels.

| Noise Level($\sigma$) | NRASAM | NRSAM | ASAM | SAM | SGD-M |
|---|---|---|---|---|---|
| 0.005 | **89.18** | 89.03 | 86.15 | 86.02 | 86.15 |
| 0.008 | **87.49** | 87.39 | 81.76 | 81.18 | 81.88 |
| 0.01 | **87.54** | 86.21 | 78.18 | 77.71 | 76.50 |
| 0.015 | **84.22** | 83.65 | 69.17 | 68.11 | 69.48 |

## 5.2 EXPERIMENTAL RESULTS OF TBG-NRASAM AND NRASAM

In this section, we conducted experiments related to NRASAM, which were performed after hyperparameter tuning using both the TBG method and the original NRASAM. We ensured that the $\mu(a, 0)$ of TBG-NRASAM is less than that of NRASAM. The relevant results are presented in the following tables. Table 5 shows the accuracy of five different models using TBG-NRASAM and NRASAM across three datasets under Gaussian noise with $\sigma$ set to 0.005. It is evident that the accuracy of NRASAM guided by TBG is higher. This also indicates that in a noisy environment, TBG-NRASAM has stronger robustness and generalization capabilities compared to traditional SAM and its variants.

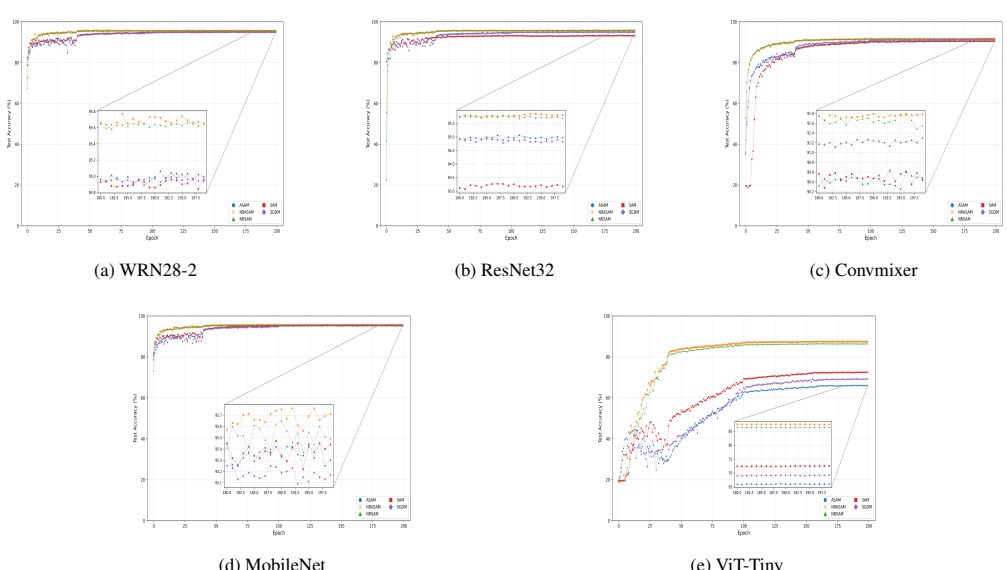

(a) WRN28-2          (b) ResNet32          (c) Convmixer

(d) MobileNet          (e) ViT-Tiny

Figure 5: Performance of five models with Gaussian noise $N(0, 0.005)$ on SVHN.

Table 5: Comparison of test accuracy between NRASAM and TBG-NRASAM on different datasets

| Dataset | Optimizer | Wrn28-2 | ResNet32 | ViT Tiny | MobileNet |
|---|---|---|---|---|---|
| CIFAR-10 | NRASAM | 90.45 | 88.75 | 75.26 | 89.50 |
| CIFAR-10 | TBG-NRASAM | **90.82** | **88.97** | **75.83** | **89.62** |
| CIFAR-100 | NRASAM | 67.36 | 63.28 | 47.53 | 66.93 |
| CIFAR-100 | TBG-NRASAM | **67.39** | **63.48** | **48.63** | **67.27** |
| SVHN | NRASAM | 95.82 | 95.86 | 87.51 | 95.78 |
| SVHN | TBG-NRASAM | **95.99** | **95.89** | **88.46** | **95.80** |

## 6 CONLUSION

This paper proposes a TBG and a boundedness theorem based on discrete time system theory, and develops a noise resistant adaptive sharpness minimization method (TBG-NRASAM). It accelerates convergence and effectively determines the upper bound. By establishing the $\mu$ - contraction theory, it provides a unified boundedness analysis tool for non - exponentially decaying systems and rigorously proves the algorithm's convergence and noise-resistance. Experimental results show that this method significantly outperforms existing SAM-based algorithms in various architectures and benchmark tests. Especially in complex scenarios with label noise and data perturbations, its advantages in enhancing model generalization and stability are more prominent. This study provides a new theoretical perspective and effective tool for optimization problems seeking flat minima in noisy environments. The proposed TBG framework is expected to be extended to fields with strict privacy requirements, such as federated learning, to further explore relevant solutions for better generalization.

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

# A APPENDIX

## A.1 SOME LEMMA

**Lemma 1.** *(Jr. et al. (2023)) If $f''$ is continuous and $M$ is any upper bound for the values of $\| f'' \|$ on $[a, b]$, then the error $E_M$ in the approximation of the integral of $f$ from $a$ to $b$ for $n$ steps satisfies the inequality*

$$|E_M| \leq \frac{M(b-a)^3}{24n^2}.$$

**Lemma 2.** *(Horn & Johnson (2012)) Let $A \in \mathbb{M}_n$ and $\varepsilon > 0$ be given. There exists a constant $C = C(A, \varepsilon)$ such that*

$$|(A^k)_{ij}| \leq C \left( \rho(A) + \varepsilon \right)^k$$

*for all $k \in N$ and $i, j \in P_N$.*

## A.2 PROOF OF THEOREM 1

*Proof.* It can be seen from the previous definition of the linear system that

$$x_m = A_{m-1}x_{m-1} = \mathcal{A}(m, n)x_n.$$

And according to Definition 2, there exist constants $K$ and $\alpha$ that satisfy:

$$\| \mathcal{A}(m, n) \|_2 \leq K\mu^\alpha(m, n).$$

Then, we have

$$\| x_m \|_2 \leq K\mu^\alpha(m, n)x_n.$$

The proof is complete. □

## A.3 PROOF OF THEOREM 2

*Proof.* It can be seen from the equation of the linear system with perturbation that

$$x_m = A_{m-1}x_{m-1} + f_{m-1} = \mathcal{A}(m, n)x_n + \sum_{l=n}^{m-1} \mathcal{A}(m, l+1)f_l.$$

And according to Definition 2 and the triangle inequality of norms, there exist constants $K$ and $\alpha$ that satisfy:

$$\| x_m \|_2 \leq \| \mathcal{A}(m, n) \|_2 \, x_n + \sum_{l=n}^{m-1} \| \mathcal{A}(m, l+1) \|_2 \cdot \| f_l \|_2$$

$$\leq K\mu^\alpha(m, n)x_n + \sum_{l=n}^{m-1} K\mu^\alpha(m, l+1) \cdot f_{\max}$$

$$\leq K\mu^\alpha(m, n)x_n + Kf_{\max} \sum_{l=n}^{m-1} \mu^\alpha(m, l+1).$$

We focus on the summation part of the second term:

$$\sum_{l=n}^{m-1} \mu^\alpha(m, l+1) = \mu^\alpha(m, n+1) + \mu^\alpha(m, n+2) + \cdots + \mu^\alpha(m, m)$$

$$= \mu^\alpha(m, m) + \mu^\alpha(m, m)\mu^\alpha(m, m-1)$$

$$+ \mu^\alpha(m, m)\mu^\alpha(m, m-1)\mu^\alpha(m-1, m-2) + \cdots$$

$$+ \mu^\alpha(m, m) \prod_{l=n+2}^{m} \mu^\alpha(l, l-1).$$

Note that $\mu(m,m) \cdot \mu(m,m) = \mu(m,m)$, then $\mu(m,m) = 1$. Furthermore, by Definition 1, we have $\mu(t, t-1) \leq b < 1$. Hence,

$$\sum_{l=n}^{m-1} \mu^\alpha(m, l+1) \leq 1 + b^\alpha + b^{2\alpha} + \cdots + b^{\alpha(m-n-1)}$$

$$= \frac{1 - b^{\alpha(m-n)}}{1 - b^\alpha}.$$

Then,

$$\| x_m \|_2 \leq K\mu^\alpha(m,n)x_n + Kf_{\max} \sum_{l=n}^{m-1} \mu^\alpha(m, l+1)$$

$$\leq K\mu^\alpha(m,n)x_n + \frac{Kf_{\max}(1 - b^{\alpha(m-n)})}{1 - b^\alpha}.$$

The proof is complete. $\qquad\qquad\square$

### A.4 PROOF OF THEOREM 3

*Proof.* Discretizing the integral using Euler's method will lead to error $o(\tau^2)$, i.e. $\int_0^t h_\tau \mathrm{d}\tau \approx \tau \sum_{j=0}^k h_j + o(\tau^2)$. Then, the noise-robust update rule can be rewritten:

$$\boldsymbol{w}_{k+1} = \boldsymbol{w}_k - \gamma\boldsymbol{h}_{k-1} + \delta\sum_{j=0}^k \boldsymbol{h}_j + o_1(\tau^2). \tag{3}$$

By substituting the formula of gradient descent method $\boldsymbol{w}_{k+1} = \boldsymbol{w}_k - \eta\boldsymbol{h}_k$, the equation (2) can be rewritten:

$$-\eta\boldsymbol{h}_k = -\gamma\boldsymbol{h}_{k-1} + \delta\sum_{j=0}^k \boldsymbol{h}_j + o_1(\tau^2). \tag{4}$$

Similarly, the formula for the $(k-1)$-th step is

$$-\eta\boldsymbol{h}_{k-1} = -\gamma\boldsymbol{h}_{k-2} + \delta\sum_{j=0}^{k-1} \boldsymbol{h}_j + o_2(\tau^2). \tag{5}$$

According to equations (3) and (4), we have

$$(\eta + \delta)\boldsymbol{h}_k = (\eta + \gamma)\boldsymbol{h}_{k-1} - \gamma\boldsymbol{h}_{k-2} + o_2(\tau^2) - o_1(\tau^2). \tag{6}$$

Let $o(\tau^2) = o_2(\tau^2) - o_1(\tau^2)$, then the aforementioned equation (5) can be described as

$$\upsilon_k = H\upsilon_{k-1} + o(\tau^2), \tag{7}$$

where $\upsilon_k = [h_k^{\mathrm{T}}, h_{k-1}^{\mathrm{T}}]^{\mathrm{T}}$. Matrix $H$ is defined as

$$H = \begin{bmatrix} \frac{\eta+\gamma}{\eta+\delta} & \frac{-\gamma}{\eta+\delta} \\ 1 & 0 \end{bmatrix}.$$

Due to Lemma 1 in A.1, we bound the term $o(\tau^2)$, i.e.,

$$|o(\tau^2)| \leq \frac{Mt_1\tau^2 - Mt_2\tau^2}{24} = \frac{Mk\tau^3 - M(k-1))\tau^3}{24} = \frac{M\tau^3}{24}, \tag{8}$$

where $M$ is any upper bound for the values of $\|h''\|$ on $[0, t]$.

It can be generalized from (6) that

$$\upsilon_k = H\upsilon_{k-1} + o(\tau^2) = H^{k-1}\upsilon_1 + \sum_{l=1}^{k-1} H^{k-l-1}o(\tau^2). \tag{9}$$

Since the spectral radius of the H is

$$\max\left\{\left|\frac{\eta + \gamma + \sqrt{(\eta - \gamma)^2 - 4\gamma\delta}}{2(\delta + \eta)}\right|, \left|\frac{\eta + \gamma - \sqrt{(\eta - \gamma)^2 - 4\gamma\delta}}{2(\delta + \eta)}\right|\right\} = \rho.$$

By selecting an appropriate $\epsilon > 0$ such that the inequality $\rho < \beta \triangleq \rho + \epsilon < \mu(a, 0) < 1$ is satisfied. According to Lemma 2, there exist $C_\beta > 0$ such that

$$|(H^k)_{ij}| \le C_\beta \beta^k,$$

for all $k \in N$ and $i, j = 1, 2$.

Since the Frobenius norm of a matrix is always greater than or equal to its 2-norm, the following inequality holds for all $k \in N$:

$$\|H^k\|_2 \le \sqrt{\sum_{i,j}|(H^k)_{ij}|^2} \le \sqrt{2^2 \cdot [C_\beta \beta^k]^2} = 2C_\beta \beta^k.$$

Let $\tilde{\mu}(m, n) = \beta^{m-n}$, then it can be verified that $\tilde{\mu}(m, n)$ satisfies Definition 1. Therefore, $\|H^{k-1}\| \le \tilde{\mu}(k, 1)$ hold.

According to Theorem 2 , we have

$$\|v_k\|_2 \le 2C_\beta \tilde{\mu}(k, 1)v_1 + \frac{C_\beta M \tau^3 (1 - \mu(a, 0)^{(k-1)})}{12(1 - \mu(a, 0))}.$$

As a result, we have

$$\lim_{k\to\infty}\|v_k\|_2 \le \lim_{k\to\infty} 2C_\beta \tilde{\mu}(k, 1)v_1 + \lim_{k\to\infty} \frac{C_\beta M \tau^3 (1 - \mu(a, 0)^{(k-1)})}{12(1 - \mu(a, 0))}$$
$$\le \frac{C_\beta M \tau^3}{12(1 - \mu(a, 0))}.$$

Hence, it follows that

$$\lim_{k\to\infty}\|h_k\|_2 \le \frac{C_\beta M \tau^3}{6\sqrt{2}(1 - \mu(a, 0))}.$$

The proof is complete. $\qquad\square$

A.5   PROOF OF THEOREM 4

*Proof.* Incorporating the expression of the gradient descent update $w_{k+1} = w_k - \eta h_k$ into Equation (9) leads to the following:

$$-\eta h_k = -\gamma h_{k-1} + \delta \sum_{j=0}^{k} h_j + o_1(\tau^2) + e_k. \tag{10}$$

Similarly, the formula for the $(k-1)$-th step is

$$-\eta h_{k-1} = -\gamma h_{k-2} + \delta \sum_{j=0}^{k-1} h_j + o_2(\tau^2) + e_{k-1}. \tag{11}$$

From the preceding equations (10) and (11), it follows that

$$(\eta + \delta)h_k = (\eta + \gamma)h_{k-1} - \gamma h_{k-2} + o(\tau^2) - (e_k - e_{k-1}). \tag{12}$$

Let $o(\tau^2) = o_2(\tau^2) - o_1(\tau^2)$ and $n_k = [e_k^T - e_{k-1}^T, 0]^T$, then equation (12) can be transformed into

$$v_k = Hv_{k-1} + o(\tau^2) - n_k, \tag{13}$$

where the definition of $H$ and $v_k$ is the same as Theorem 1.

Hence,

$$v_k = H^{k-1}v_1 + \sum_{l=1}^{k-1} H^{k-l-1}(o(\tau^2) - n_k).$$

Let $2d$ be the dimension of $n_i$ and $c_i$ be the element of the maximum value among $n_i$. Thus, we obtain

$$\| n_k \|_2 \leq \max_{2 \leq i \leq k} \| n_i \|_2 \leq \sqrt{2d} \sup_{2 \leq i \leq k} |c_i|.$$

Combining equation (7), we have the following:

$$\| o(\tau^2) - n_k \|_2 \leq \frac{M\tau^3}{24} + \sqrt{2d} \sup_{2 \leq i \leq k} |c_i|.$$

According to Lemma 1, we derive the upper bound of the 2-norm of $v_k$:

$$\|v_k\|_2 \leq 2C_\beta \tilde{\mu}(k,1)v_1 + \frac{2C_\beta(1 - \mu(a,0)^{(k-1)})}{(1 - \mu(a,0))}\left(\frac{M\tau^3}{24} + \sqrt{2d} \sup_{2 \leq i \leq k} |c_i|\right).$$

As a result, we have

$$\lim_{k \to \infty} \|v_k\|_2 \leq \lim_{k \to \infty} 2C_\beta \left\{ \tilde{\mu}(k,1)v_1 + \frac{(1 - \mu(a,0)^{(k-1)})}{(1 - \mu(a,0))}\left(\frac{M\tau^3}{24} + \sqrt{2d} \sup_{2 \leq i \leq k} |c_i|\right)\right\}$$

$$\leq \frac{2C_\beta}{(1 - \mu(a,0))}\left(\frac{M\tau^3}{24} + \sqrt{2d} \sup_{2 \leq i \leq k} |c_i|\right).$$

Consequently, we have

$$\lim_{k \to \infty} \|h_k\|_2 \leq \frac{\sqrt{2}C_\beta}{(1 - \mu(a,0))}\left(\frac{M\tau^3}{24} + \sqrt{2d} \sup_{2 \leq i \leq k} |c_i|\right).$$

The proof is complete. $\qquad\square$

