# OpenReview forum: "TBG-Driven Minimization of Noise-Resistant Adaptive Sharpness Awareness"
_ICLR.cc/2026/Conference — Submitted to ICLR 2026_

### Official Review · Reviewer_JY2Z · 2025-10-27

**Soundness:** 2
**Presentation:** 1
**Contribution:** 2
**Rating:** 0
**Confidence:** 3

**Summary:**

The paper is poorly written, with many arbitrarily defined names and techniques. The overall flow is difficult to follow for beginners, and the exposition lacks clarity and rigor. While more mature readers can roughly infer the authors’ intended ideas, the lack of precision and structure is already a red flag for a venue such as ICLR.

**Strengths:**

While the topic may be of potential interest, I did not identify any clear technical or empirical strengths in the current version of the paper. This is mainly because the presentation, logical flow, and definitions are unclear, making it difficult to discern the intended contributions. I generally avoid evaluating a paper based on assumptions or guesswork about the authors’ intent.

**Weaknesses:**

W1. The paper motivates the addition of noise to gradients from a privacy perspective in the Introduction, and adds Gaussian noise to gradients in experiment. However, the proposed algorithm does not actually address privacy concerns. It neither employs standard differential privacy (DP) techniques nor provides any formal privacy guarantees. This leaves adding noise to gradients not well-justified.
The authors should consider presenting more convincing scenarios or motivations where gradient noise naturally arises during training to better justify their experimental setup.

W2. It is unclear what NRSAM refers to in the experiments. This algorithm does not appear to be introduced or defined anywhere in the paper. Similarly, TBG-NRASAM is mentioned in the first paragraph of Section 5 without explanation. The authors should clearly define these methods and describe how they relate to the proposed approach. Some experiments have TBG-NRASAM, and some not.

W3. The presentation does not clearly explain how TBG is applied to NRASAM or why this combination is expected to improve performance. A more detailed and structured description of the integration mechanism and its intended benefits would greatly improve the clarity of this section.

W4. The theoretical analysis is difficult to follow, as the problem setting, assumptions, and algorithms are not clearly or consistently explained. The lack of precise definitions and organization makes it challenging to assess the correctness and significance of the theoretical results.

**Questions:**

I believe that a thorough revision and careful polishing of the writing, organization, and technical exposition could substantially improve the overall quality and readability of the paper.

---

### Official Review · Reviewer_dccB · 2025-10-30

**Soundness:** 2
**Presentation:** 1
**Contribution:** 2
**Rating:** 2
**Confidence:** 4

**Summary:**

The authors propose a new optimization method, TBG-NRASAM, which is reported to be more robust to Gaussian noise than SAM. In particular, leveraging the time-based generator theorem, they reuse the gradient of the previous weights to enhance robustness against Gaussian perturbations. The experimental results appear to support the effectiveness of the proposed algorithm under Gaussian noise.

**Strengths:**

The idea of combining a time-based generator with sharpness-aware minimization is interesting and, to the best of my knowledge, has not been explicitly explored in prior work. Moreover, it is noteworthy that the proposed method enhances robustness to Gaussian noise while relying solely on previous gradients. This direction could be further investigated from the broader perspective of robustness in AI models.

**Weaknesses:**

(Major)
There are several issues that should be addressed before publication.

Sections 2 (“Sharpness-Aware Minimization”) and 3 (“Time-Based Generator”) should be merged into a preliminaries section. These sections essentially introduce existing methods and concepts, but in the current form, they are presented as if they were the authors’ contributions. Important related work is also missing from these sections, which may mislead readers into thinking that the algorithms described there are original to this paper. The authors should clearly separate (i) background / existing techniques and (ii) their own contribution.

The main concern about the proposed method is that the paper mostly emphasizes performance under noisy (Gaussian) settings, while giving less attention to performance on clean examples. Even if the goal is robustness, a practically useful optimizer should not severely degrade performance on clean data. In Table 5, the authors should discuss in more detail the performance of baselines such as SAM, ASAM, and SGD-M, not only under noise but also on clean data.
In addition, several recent variants and analyses of SAM should at least be cited and, if possible, included as comparison methods:
- Ji, Jie, et al. “A single-step, sharpness-aware minimization is all you need to achieve efficient and accurate sparse training.” NeurIPS 37 (2024): 44269–44290.
- Li, Bingcong, Liang Zhang, and Niao He. “Implicit regularization of sharpness-aware minimization for scale-invariant problems.” NeurIPS 37 (2024): 44444–44478.
- Mueller, Maximilian, et al. “Normalization layers are all that sharpness-aware minimization needs.” NeurIPS 36 (2023): 69228–69252.
Since the proposed method builds on the SAM family, omitting these works weakens the contribution.


Furthermore, in Algorithm 1, the proposed method uses the gradient of $L_B(w_{t-1}^{adv})$ and the gradient of $L_B(w^{adv}_{t})$. It seems necessary to use the same mini-batch across these steps in each iteration. However, the paper does not explain this implementation detail. If the same batch is used, please state this explicitly and discuss the computational implications. If different batches were actually used in the experiments, then the implemented method is not exactly the one described in the algorithm, and this discrepancy should be explained.

While the proposed algorithm shows better performance under random Gaussian noise, the paper should also demonstrate that it maintains reasonable accuracy on other types of noisy data. Authors should consider CIFAR-10-C and CIFAR-100-C. Otherwise, the improvement may come from overfitting to the Gaussian noisy setting. A discussion about this trade-off, ideally with additional tables/plots, would strengthen the paper.

(Minor)
- Figures are not easy to read; please increase their resolution and/or font sizes.
- Tables should have consistent significant figures/decimal places (e.g., Table 1). Inconsistent formatting makes it harder to compare methods.

**Questions:**

Refer to Weaknesses.

---

### Official Review · Reviewer_A6sr · 2025-11-02

**Soundness:** 2
**Presentation:** 2
**Contribution:** 3
**Rating:** 4
**Confidence:** 2

**Summary:**

This paper addresses the problem of degraded model generalization under noisy training scenarios and proposes a Time-Base Generator (TBG) and noise-resistant adaptive sharpness-aware minimization methods (NRASAM, TBG-NRASAM). The key contributions are: proposing a TBG based on discrete systems and its boundedness theorem, providing a stability analysis tool for non-exponentially decaying systems; designing NRASAM, which integrates adaptive sharpness measurement and historical gradient integration mechanism to simultaneously suppress intrinsic parameter scaling interference and extrinsic noise perturbation; optimizing parameters via TBG to propose TBG-NRASAM, improving algorithm convergence. Theoretical proofs verify the algorithm's convergence and noise resistance, and experiments show that the method outperforms SAM-series methods across multiple datasets, model architectures, and noise intensities, providing an effective optimization solution for noisy training scenarios such as differential privacy protection and federated learning.

**Strengths:**

1.The theoretical derivation is rigorous, and the algorithm's convergence and noise resistance are fully verified through 4 theorems; the experimental design is comprehensive, covering multiple datasets, model architectures, and noise intensities. Ablation experiments validate the effectiveness of TBG parameter adjustment, with highly credible conclusions.
2. The paper has a coherent structure, progressing layer by layer from problem formulation, theoretical foundation, algorithm design to experimental verification; the formula derivation is detailed, the appendix supplements complete theorem proofs, and figures intuitively show the algorithm's convergence process and performance advantages, facilitating reader understanding.
3. Focusing on the core pain points of practical scenarios such as differential privacy and federated learning, the proposed method can be directly integrated into existing optimization frameworks, which is of great practical significance for model training requiring noise protection.

**Weaknesses:**

1. Limited coverage of noise types: Experiments only verify the anti-interference performance of Gaussian gradient noise, not involving common noise types in practical scenarios such as label noise and data input noise; the performance boundary of the algorithm under extreme noise intensities (e.g., σ>0.02) is not evaluated.
2. Lack of computational overhead analysis: NRASAM introduces a historical gradient integration mechanism, which may increase additional computational and storage overhead. However, the paper does not compare its parameter count, FLOPs, and training/inference time with SAM and ASAM, lacking efficiency-performance trade-off analysis.

**Questions:**

1. What is the performance of the algorithm under other common noise types such as label noise and data input noise? Can supplementary experiments be provided to verify its generalization ability in diverse noise scenarios?
2. Does the historical gradient integration mechanism of NRASAM increase computational and storage overhead?

---

### Official Review · Reviewer_7hfA · 2025-11-04

**Soundness:** 2
**Presentation:** 1
**Contribution:** 1
**Rating:** 2
**Confidence:** 4

**Summary:**

- This paper proposes noise-resistant adaptive sharpness-awareness minimization (NRASAM) variants.
- The proposed method is based on the time-based generator and a boundedness theorem in a discrete system.
- The authors seem to conduct image classification experiments to demonstrate the superiority of their proposed method.

**Strengths:**

- The proposed methods are theoretically grounded.

**Weaknesses:**

- The paper needs to be revised for a better understanding of the readers.
- As far as I understand, the proposed NRASAM is a combination of ASAM and NRSAM.
- It is hard to intuitively understand the concept of the discrete system and the proposed TBG-NRASAM. Please supplement the conceptual relation.
- No error bars in the experimental results.

**Questions:**

- Please interpret the meanings of Theorems 3 and 4. Also, in Line 267, ... the rule for updating (2) is restructured to: ... eq_num (2)? I do not understand this part. Maybe a typo?
- So, what is the proposed TBG-NRASAM? Please properly provide the algorithm.
- Please specify the difference among ASAM, NRSAM, and NRASAM, including the algorithms and the contributions as well.
- The authors proposed TBG for the discrete system, but ended up with an image classification task, right? What is the definition of a discrete system? How does it connect to the image classification models? Moreover, the experiments were the image classification experiments, right? In the paper, it just says "experiments on CIFAR-10 dataset", etc.

---

### Meta-Review · Area_Chair_T5hp · 2025-12-29

**Summary:**

This paper proposes TBG-NRASAM, a noise-resistant variant of sharpness-aware minimization motivated by noisy training scenarios. Reviewers consistently raise substantial concerns about clarity, positioning, and empirical validation. Multiple reviewers note that the presentation is poor, with unclear definitions (e.g., NRASAM, NRSAM, TBG-NRASAM), weak separation between prior work and contributions, and theoretical analysis that is difficult to follow. Empirically, the evaluation focuses narrowly on Gaussian gradient noise, lacks error bars and efficiency/overhead analysis, omits important recent SAM variants, and provides insufficient discussion of performance on clean data or other realistic noise settings.

Crucially, the authors did not provide a rebuttal to address these concerns. Given the absence of clarification or additional evidence, I recommend rejection.

**Reviewer Concerns:**

The authors did not provide a rebuttal to address these concerns.

**Reviewer Scores:**

0,2,2,4

---

### Decision · Program_Chairs · 2026-01-26

Reject